# Novel Genetic Diagnoses in Septo-Optic Dysplasia

**DOI:** 10.3390/genes13071165

**Published:** 2022-06-28

**Authors:** Linda M. Reis, Sarah Seese, Mohit Maheshwari, Donald Basel, LuAnn Weik, Julie McCarrier, Elena V. Semina

**Affiliations:** 1Department of Pediatrics and Children’s Research Institute, Medical College of Wisconsin and Children’s Wisconsin, Milwaukee, WI 53226, USA; lreis@mcw.edu (L.M.R.); sseese@mcw.edu (S.S.); dbasel@mcw.edu (D.B.); lweik@mcw.edu (L.W.); jmccarrier@mcw.edu (J.M.); 2Department of Pediatric Radiology, Medical College of Wisconsin and Children’s Wisconsin, Milwaukee, WI 53226, USA; mmaheshwari@chw.org; 3Department of Genome Sciences, University of Washington, Seattle, WA 98195, USA; 4Departments of Ophthalmology and Cell Biology, Medical College of Wisconsin, Milwaukee, WI 53226, USA

**Keywords:** septo-optic dysplasia, *SHH*, *ARID1A*, *SOX2*

## Abstract

Septo-optic dysplasia (SOD) is a developmental phenotype characterized by midline neuroradiological anomalies, optic nerve hypoplasia, and pituitary anomalies, with a high degree of variability and additional systemic anomalies present in some cases. While disruption of several transcription factors has been identified in SOD cohorts, most cases lack a genetic diagnosis, with multifactorial risk factors being thought to play a role. Exome sequencing in a cohort of families with a clinical diagnosis of SOD identified a genetic diagnosis in 3/6 families, de novo variants in *SOX2*, *SHH*, and *ARID1A*, and explored variants of uncertain significance in the remaining three. The outcome of this study suggests that investigation for a genetic etiology is warranted in individuals with SOD, particularly in the presence of additional syndromic anomalies and when born to older, multigravida mothers. The identification of causative variants in *SHH* and *ARID1A* further expands the phenotypic spectra associated with these genes and reveals novel pathways to explore in septo-optic dysplasia.

## 1. Introduction

Septo-optic dysplasia (SOD) is a rare, heterogenous phenotype reported in 1 out of 10,000 live births and is characterized by any combination of midline neuroradiological anomalies, including absence of the septum pellucidum or agenesis of the corpus callosum, hypoplasia of the optic chiasma/nerves, and hypothalamic–pituitary dysfunction [1]. The phenotype is highly variable, with additional anomalies present in many cases. Several transcription factors were shown to play a role in SOD, with the first factor, *HESX1*, identified in 1998 [2] and variants in *SOX3*, *SOX2*, and *OTX2* being more recently implicated [1,3]. However, variants in these genes explain only a small number of cases, highlighting the need to identify additional novel players. In the majority of cases, SOD is sporadic with a low recurrence risk; young maternal age, primigravida status, viral infections, and other environmental causes have also been implicated, suggesting that the phenotype is likely to be multifactorial in many cases [1,3].

## 2. Materials and Methods

This human study was conducted according to the guidelines of the Declaration of Helsinki and approved by the Institutional Review Board of Children’s Wisconsin. Exome sequencing was undertaken through Psomagen (previously Axeq; Rockville, MD, USA), the University of Washington Center for Mendelian Genomics, or via clinical testing and then analyzed, as previously described, utilizing the SNP &Variation Suite or VarSeq software (Golden Helix, Bozeman, MT, USA), including annotations for gnomAD v2.1.1, OMIM genes, CADD Scores 1.4, and REVEL Functional Predictions [4]. The data were reviewed for rare variants in known SOD genes (*HESX1*, *SOX3*, *SOX2*, *OTX2*) and OMIM genes, along with a standard trio analysis (when available) and review of ultra-rare (≤5 alleles in gnomAD) and damaging (Loss of Function (LOF) or REVEL > 0.4 and CADD > 20) coding variants. Causative variants were confirmed by Sanger sequencing using region-specific primers. Six families with a clinical diagnosis of SOD in the proband underwent exome sequencing, including four trios (affected proband plus unaffected parents), one quad (affected siblings plus unaffected parents), and one singleton. Tissue-expression patterns for novel genes were investigated in The Human Protein Atlas (proteinatlas.org).

## 3. Results

Causative variants were identified in three SOD families, including one variant in *SOX2*. No rare variants were identified in other SOD genes. 

In Family 1, the affected individual is a 5-year-old White male with mild optic nerve hypoplasia, absent septum pellucidum, hypoplasia of corpus callosum, and dilated lateral ventricles identified by Brain MRI; the individual was born as the second child to a 30-year-old mother (Figure 1). He has global developmental delay, hypotonia with unsteady gait requiring the use of supports (forearm crutches), complex seizures, and hyperopia with mild optic disc pallor by clinical exam. Trio exome sequencing identified a de novo variant in *SOX2*, NM_003106.4:c.70_89del20 p.(Asn24Argfs*65); Sanger sequencing confirmed the presence of the variant in the child and its absence in the mother (Table 1; Figure 2).

In Family 2, the affected individual is an 18-month-old Hispanic female with absent septum pellucidum, partially absent corpus callosum (anterior absent), left optic nerve hypoplasia, and right retinal coloboma with mild asymmetry of the orbits noted by clinical exam, with the right eye appearing somewhat smaller than the left, born as the third child of a 35-year-old mother. She also has a bilateral complete cleft palate and a right complete cleft lip, a single central incisor, microcephaly, mandibular hypoplasia, alveolar gap, global developmental delay, failure to thrive (1st centile), asymmetric thigh creases, coxa valga, dysphagia requiring g-tube feeding, and diabetes insipidus due to pituitary dysfunction. At 13 months of age, measurements confirmed normal placement of the eyes with an inner canthal distance of 2 cm (50th centile) and an outer canthal distance of 7 cm (25th–50th centile). Trio exome sequencing identified a de novo variant in *SHH*, NM_000193.2:c.562+1G>A, which was confirmed to be present in the child and absent in both parents by Sanger sequencing (Figure 2).

In Family 3, the affected individual is a 6-week-old White male with absent septum pellucidum, absent corpus callosum, ventriculomegaly, aqueductal stenosis, and intraventricular hemorrhage identified on Brain MRI (Figure 1), born at 33 weeks of gestation as the 5th child of a 32-year-old mother. Eye exam at 5 weeks of age identified bilateral optic disc pallor, asymmetric optic discs with the right larger than the left, and a peripapillary halo also more notable on the right. Additional systemic anomalies included a ventricular septal defect and a patent foramen ovale, 13 pairs of ribs, bilateral clinodactyly, single palmar crease, broad large toe with hypoplastic nail, cleft palate, choanal atresia, seizures, apnea, and dysmorphic facial features, including down-slanting palpebral fissures, long columella, low-set and posteriorly rotated ears, depressed nasal bridge, scant hair due to premature birth; he died at 6 weeks of age. Trio exome sequencing identified a de novo variant in *ARID1A*, NM_006015.6:c.6625C>T p.(Gln2209*), which was confirmed to be present in the child and absent in both parents by Sanger sequencing (Figure 2). The variant appeared to be mosaic: it was present in 59/179 (33%) exome reads and had a lower peak as determined by Sanger sequencing (Figure 2).

Family 4 consists of a 4-year-old White/Native Hawaiian male with hypoplastic corpus callosum and genu, bilateral optic nerve hypoplasia, and panhypopituitarism with an ectopic posterior and a severely hypoplastic anterior pituitary gland on Brain MRI (Figure 1) along with global delay and seizure-like activity, gastroschisis with jejunal atresia, and a ventricular septal defect, born to a 23-year-old primigravida mother. He has a 2-year-old affected brother with bilateral optic nerve hypoplasia, a mildly hypoplastic corpus callosum, and panhypopituitarism with posterior pituitary ectopia with small sella and hypoplasia of the anterior pituitary gland on Brain MRI (Figure 1) along with global delay, depressed nasal bridge, and simple, cupped ears. Quad exome sequencing did not identify a causative variant and no rare variants were identified in known SOD genes; variants of uncertain significance included compound heterozygous variants in *MIB2* and a hemizygous missense variant in *AKAP4* (Table 2). A review of the shared, ultra-rare, damaging variants identified seven inherited variants of uncertain significance shared by both brothers (Appendix A), including a heterozygous missense variant in *CHD5*.

Family 5 consists of a 5-month-old White male with absent septum pellucidum, hypoplastic corpus collosum, and bilateral optic nerve hypoplasia on Brain MRI (Figure 1) along with growth hormone deficiency, borderline micropenis, and mild global developmental delay, born to a 29-year-old primigravida mother. Trio exome sequencing did not identify a causative variant; variants of uncertain significance include compound heterozygous missense variants in *FAT3* and a homozygous frameshift variant in *RPTN* (Table 2). Review of ultra-rare damaging variants identified seven inherited variants of uncertain significance (Appendix A), including a missense variant in *TRPM3*.

Family 6 consists of a 7-year-old Hispanic female with a hypoplastic corpus callosum and midline anomalies as well as bilateral optic nerve hypoplasia on Brain MRI (Figure 2) along with highly arched palate, spastic hemiplegia, dysphagia, enamel hypoplasia, high-arched palate, hemolytic anemia, and global delay, born to a 20-year-old primigravida mother. Singleton exome sequencing did not identify a causative variant. A review of ultra-rare damaging variants identified 17 variants of uncertain significance (Appendix A), including a homozygous variant in *DMXL1* and a heterozygous variant in *CCDC13* (Table 2).

## 4. Discussion

While septo-optic dysplasia (SOD) typically has a low rate of genetic diagnosis, the genetic analysis of this small cohort of individuals with SOD identified a genetic etiology in 50% of the families. Cases with a genetic diagnosis were more likely to have atypical optic nerve findings, but all had been given a clinical diagnosis of septo-optic dysplasia prior to genetic testing, highlighting the variability of this phenotype. Interestingly, cases without a genetic diagnosis were more likely to be born to younger, primigravida mothers, consistent with previous associations [1,3], although potentially contributing variants were identified in novel genes in each of these families. 

*SOX2* intragenic variants and deletions are the most common cause of anophthalmia/microphthalmia, typically syndromic with commonly seen esophageal, genitourinary, and neurological anomalies [5]. A connection to SOD was noted in a mouse model, and subsequent screening in a cohort of individuals with *SOX2* variants identified pituitary hypoplasia and hypogonadotropic hypogonadism along with anomalies of the corpus callosum and medial temporal structures [6]. As expected, all of the individuals with loss-of-function variants also had anophthalmia/microphthalmia, with variable additional syndromic features. Absent septum pellucidum was only noted in two individuals with missense variants (p.(Gly130Ala) and p.(Ala191Thr)) and an isolated SOD phenotype with normal eye size; both of these variants were inherited from phenotypically normal parents and are now known to be present in the general population, with population max frequencies of 0.02% and 0.05% in gnomAD, higher than the frequency of SOD, suggesting that these are likely to be benign, population-specific variants. Individual 1 of this study is the first case of SOD with normal eye size and a lack of additional birth defects to have a loss-of-function variant in *SOX2*. Interestingly, the identified *SOX2* variant, c.70_89del, is a recurrent variant now reported in 20 individuals; while the majority of cases presented with a severe phenotype of syndromic anophthalmia/microphthalmia, phenotypic variability has been reported in some cases [5]. An abnormal gait, often described as ataxic and requiring the use of a walker or other assistive devices, is typical for *SOX2* disruption [5].

In humans, *SHH* variants are associated with holoprosencephaly (HPE), another developmental anomaly affecting the brain [7], and explain up to 37% of dominant HPE [8]. HPE is characterized by incomplete separation of the forebrain into right and left hemispheres, typically associated with craniofacial anomalies including microcephaly, hypotelorism, single central incisor, and cleft lip/palate [8]. Phenotypic variability is well-recognized for *SHH,* with family members often presenting with only subtle midline craniofacial features or developmental delays/ADHD [9,10]. While the conditional knockout of *Shh* in the hypothalamus of mice resulted in an SOD phenotype [11], this is the first association of variants of this gene with an SOD diagnosis in humans. The specific variant identified in Family 2 was previously reported in two cases with holoprosencephaly [12,13]. The presence of a cleft lip/palate along with SOD may suggest *SHH* disruption. Interestingly, Sox2 and Shh act in the same pathway, with Sox2/3 expression being required for *Shh* expression in the hypothalamus (via direct action of a long-range *Shh* enhancer) and disruption of this pathway in the hypothalamus was found to result in SOD in mice [11]. Another major SOD gene, the paired homeodomain transcription factor *HESX1*, is also implicated in the Sox2 pathway, with SOX2 binding to the *Hesx1* promoter in vitro; observations of decreased expression of *Hesx1* in *Sox2*-deficient mice indicate likely direct regulation of *Hesx1* by Sox2 [6]. 

*ARID1A* encodes a member of the SWItch/Sucrose NonFermenting (SWI/SNF) complex; variants in genes encoding the components of the SWI/SNF complex result in Coffin-Siris syndrome, a syndromic form of intellectual disability frequently associated with agenesis or hypoplasia of the corpus callosum [14,15]. While SOD has not been reported, other syndromic features seen in this individual show a strong overlap and early lethality has been seen with *ARID1A* variants in particular [16]. The variant identified here, c.6625C>T p.(Gln2209*), is the most C-terminal variant identified to date, but five other premature termination alleles within this final exon have been reported (HGMD [17]). The presence of multiple additional syndromic anomalies—particularly a hypoplastic big toenail, sparse hair, and heart defects—in an individual with SOD may indicate the presence of Coffin–Siris syndrome. The identification of a role for *ARID1A* in SOD proposes the involvement of a novel pathway in this disorder. Examination of *ARID1A* and related factors in SOD is warranted. 

With regard to the latter finding, it is interesting to note the identification of compound heterozygous variants of uncertain significance in *MIB2* shared by the two affected siblings in Family 4. *MIB2* (skeletrophin) is a RING finger-dependent ubiquitin ligase first identified in a screen for genes that were upregulated by truncated ARID1A (SWI1) in neuroblastoma cells displaying increased cell–cell adhesions and aggregations. MIB2 was also found to bind JAG2, a ligand in the Notch family [18,19]; Notch and Hedgehog are major signaling pathways that regulate the early steps of pituitary organogenesis and eye development, with interplay between these pathways including the restriction of *Jag2* expression by Shh [20]. A mouse model of *Mib2* deficiency showed variable neural tube closure defects [21], and an abnormal eye morphology was reported in the Mouse Genome Informatics database (http://www.informatics.jax.org/, accessed on 18 April 2022). Since the nonsense variant affects only a single transcript (out of several known isoforms) and the missense variant has weak predictions, the significance of these variants is unclear. The brothers also shared a heterozygous missense variant in *CDH5*, whose knockdown has been associated with reduced head and eye size in zebrafish [22]; however, this variant was inherited from the unaffected father.

The compound heterozygous missense variants in atypical cadherin 3, *FAT3,* that were identified in Family 5 both fall within identified domains, the Cadherin 30 and Laminin G-like domains [23]. *Fat3* has been shown to be strongly expressed in the embryonic but not adult brain in mice and rats [24], making it an interesting candidate for SOD. Variants in other members of this family, *FAT1* and *FAT2*, were found to be associated with recessive syndromic ocular coloboma [25] and dominant spinocerebellar ataxia-45 [26]. The heterozygous missense variant in *TRPM3*, with links to ocular development and intellectual disability [27,28], is also notable, but it was inherited from the unaffected father.

In Family 6, the most interesting candidate variant is a homozygous missense variant in *DMXL1*, which encodes a WD-repeat protein. This gene was identified as a candidate gene within the 5q22.3q23.3 deletion region in a patient with iris coloboma and Chiari I malformation [29]; a homozygous frameshift variant was reported in an individual with global delay, seizures, hypotonia, and optic disc edema, along with heart and kidney defects [30]. Abnormal development was also noted in a Drosophila mutant [31]. A heterozygous nonsense variant in the Coiled-Coil domain containing 13 gene, *CCDC13,* is also notable for its high CADD score (40) and absence in gnomAD. While little is known about the gene beyond a possible role in ciliogenesis [32], its RNA expression was found to be enriched in human brain and eye tissues [33]. 

The outcome of this study suggests that investigation for a genetic etiology is warranted in individuals with a clinical diagnosis of SOD, particularly in the presence of additional syndromic anomalies and when born to older, multigravida mothers. The identification of causative variants in *SHH* and *ARID1A* further expands the phenotypic spectra associated with these genes and identifies novel pathways to explore in septo-optic dysplasia.

## Figures and Tables

**Figure 1 genes-13-01165-f001:**
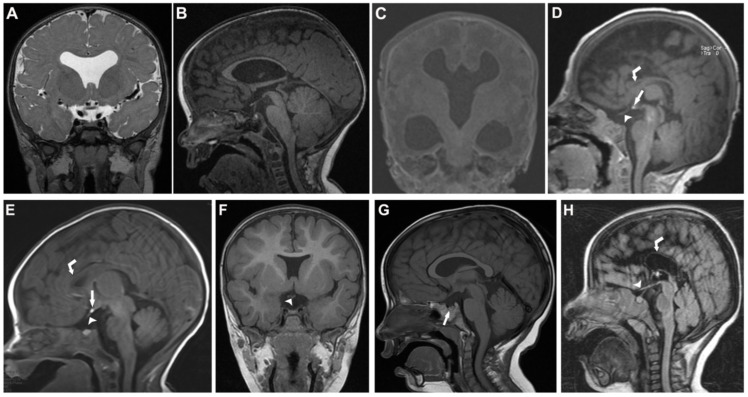
Magnetic resonance imaging (MRI) data for affected individuals. Family 1 (**A**,**B**) Coronal T2 image (**A**) from the proband showing absent septum pellucidum; Sagittal T1 image (**B**) showing hypoplasia of the corpus callosum. Family 3 (**C**) Coronal T1 image of the proband showing absent septum pellucidum, absent corpus callosum, and ventriculomegaly. Family 4 (**D**,**E**) Sagittal T1 image of proband (**D**) showing hypoplastic corpus callosum (curved white arrow), ectopic posterior pituitary (straight white arrow), and hypoplastic optic chiasm (arrowhead); Sagittal T1 image of affected brother (**E**) demonstrating a slightly hypoplastic corpus callosum (curved white arrow), ectopic posterior pituitary (white arrow), and hypoplastic optic chiasm (arrowhead). Family 5 (**F**,**G**) Coronal T1 image (**F**) with absent septum pellucidum and a small hypoplastic optic chiasm (arrowhead) (**F**); Sagittal T1 image (**G**) showing hypoplastic small pituitary (arrow). Family 6 (**H**) Sagittal T1 image showing marked hypoplasia of corpus callosum (curved white arrow) and hypoplastic optic chiasm (arrowhead).

**Figure 2 genes-13-01165-f002:**
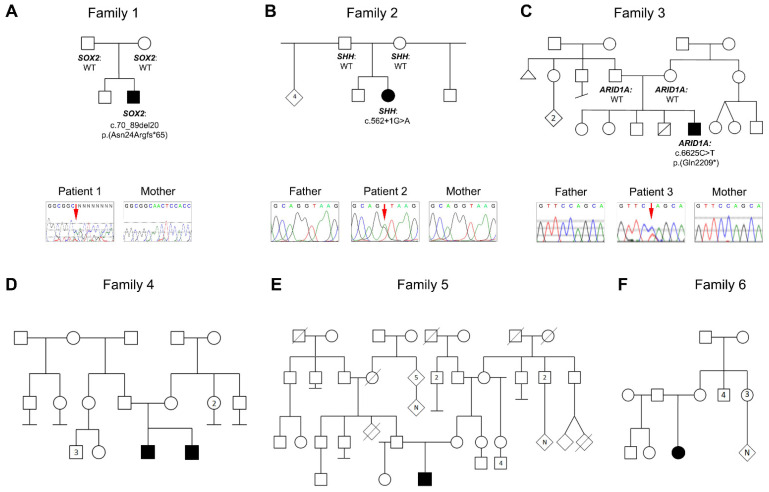
Pedigrees of Families 1–6 with a clinical diagnosis of SOD. (**A**–**C**) Pedigrees of genetically explained families with genotypes indicated for corresponding variants in *SOX2* (Family 1), *SHH* (Family 2), and *ARID1A* (Family 3) and Sanger traces showing variants in each affected individual as well as absence in unaffected parent(s). (**D**–**F**) Pedigrees of genetically unexplained families. Filled symbols represent affected individuals; empty symbols represent unaffected individuals.

**Table 1 genes-13-01165-t001:** Pathogenic variants identified in individuals with a clinical diagnosis of SOD.

Family	Gene	Nucleotide Change	Predicted Effect	MAF ^1^	ACMG/AMP Classification	Family History
1	*SOX2*	NM_003106.4: c.70_89del20	p.(Asn24Argfs*65)	NP	Pathogenic (PVS1, PS2, PM2, PP5)	De Novo
2	*SHH*	NM_000193.2: c.562+1G>A	Abnormal splicing	NP	Pathogenic (PVS1, PS2, PM2, PP5)	De Novo
3	*ARID1A*	NM_006015.6: c.6625C>T	p.(Gln2209*)	NP	Pathogenic (PVS1, PS2, PM2)	De Novo

^1^ Frequency in gnomAD v2.1.1; NP, not present.

**Table 2 genes-13-01165-t002:** Select variants of uncertain significance discovered in individuals with a clinical diagnosis of SOD.

Family	Gene	Zygosity	Nucleotide Change	Predicted Effect	MAF ^1^	CADD/ REVEL^3^		Segregation
4	*MIB2*	Compound heterozygous	NM_080875.3:c.-48C>T NM_001170688.1:c.124C>T	? p.(Arg42*)	4/185998	33	N/A	Paternal (het)
4	*MIB2*	Compound heterozygous	NM_080875.3:c.1766A>G	p.(Gln589Arg	270/ 259488	10.19	0.041	Maternal (het)
4	*AKAP4*	Hemizygous	NM_003886.3:c.1835G>A	p.(Cys612Tyr)	1/183267, 0 hemi	14.58	0.089	Maternal (het)
4	*CHD5*	Heterozygous	NM_015557.3:c.5809G>A	p.(Gly1937Arg)	NP	24.3	0.443	Paternal (het)
5	*FAT3*	Compound heterozygous	NM_001008781.3:c.9772G>A	p.(Val3258Ile)	3/271900	24	0.131	Paternal (het)
5	*FAT3*	Compound heterozygous	NM_001008781.3:c.11546G>A	p.(Arg3849Gln)	33/ 249054	24.2	0.41	Maternal (het)
5	*RPTN*	Homozygous	NM_001122965.1:c.489delA	p.(Lys163Asnfs*48)	NP	14.88	N/A	Biparental (het)
5	*TRPM3*	Heterozygous	NM_001366145.2:c.871C>T	p.(His291Tyr)	1/31412	23.2	0.419	Paternal (het)
6	*DMXL1*	Homozygous	NM_005509.6:c.9002G>A	p.(Gly3001Glu)	1/251182	31	0.614	Unknown
6	*CCDC13*	Heterozygous	NM_144719.4:c.631C>T	p.(Gln211*)	NP	40	N/A	Unknown

^1^ Frequency in gnomAD v2.1.1; ^3^ CADDphredhg19 and REVEL scores (from dbNSFP v4.1a, accessed through Varseq). N/A, not applicable; NP, not present.

## Data Availability

There are no other data associated with this manuscript.

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
