# Peer review of "Novel Genetic Diagnoses in Septo-Optic Dysplasia"

_genes, 2022, doi:10.3390/genes13071165_

Round 1
Reviewer 1 Report
The authors are to be congratulated on their in-depth genetic analysis of these patients with SOD.
Family 1. Affected child. You mention that mother also carries SOX2 variant in results end of para 1. Please clarify as the table shows her to be unaffected.
The patient in Family 2 has complex malformations, including colobomatous micropthalmia. This reviewer would put her in this category rather than in the category of ONH. Mutations in SHH generally cause holoprosencephaly as the authors mention in the discussion, hypotelorism and rarely synophthalmia. Can the authors provide a photograph of the optic nerve in this case?
Similarly in case 3, the optic nerves are described as pale with one larger than the other. This does not make for a diagnosis of ONH. Size of the optic nerve is very hard to determine, especially in very small children and infants. Photos are needed. This patient also has a multitude of systemic malformations, atypical for associations with ONH.
I am not sure that the observation/pearl that an ataxic gate requiring the use of a walker is a clue to the presence of SOX2 variants in patients with SOD. This is only one case and this is a common occurrence in patients with neurological deficits and associated brain dysfunction and malformations.
In summary, while the pathogenicity of the sequence variants has been well supported, the phenotypes they are associated with have in common some neurological malformations such as absence of the septum pellucid or corpus callosum. This reviewer is not convinced that the term SOD can be applied because of the lack of ophthalmoscopic evidence that they have ONH. Maybe the text could be rewritten acknowledging the variability of the phenotypes (as expected) and the difficulty in diagnosing classical SOD. Photographs of the optic nerves would add significantly to the paper.
Author Response
Reviewer 1
The authors are to be congratulated on their in-depth genetic analysis of these patients with SOD.
Response: Thank you!
Family 1. Affected child. You mention that mother also carries SOX2 variant in results end of para 1. Please clarify as the table shows her to be unaffected.
Response: We apologize for the confusion- we meant to say that Sanger confirmed the presence of the variant in the child and absence in the mother. This has been clarified in the text.
The patient in Family 2 has complex malformations, including colobomatous micropthalmia. This reviewer would put her in this category rather than in the category of ONH. Mutations in SHH generally cause holoprosencephaly as the authors mention in the discussion, hypotelorism and rarely synophthalmia. Can the authors provide a photograph of the optic nerve in this case?
Response: Unfortunately, this patient has been lost to follow-up and no clinical images are available. We re-reviewed the clinic note and found documentation of normal eye positioning (added to text) and clarified other ocular features. The phenotype was highly asymmetric with one eye affected with ONH only and the other eye affected with coloboma and noted to ‘appear smaller than the left’ but without a diagnosis of microphthalmia. These details were added to the revised text. Based on the presence of midline anomalies and pituitary dysfunction, this patient would meet the diagnostic criteria of having two of the three cardinal SOD features. The Genetics team following this patient gave a clinical diagnosis of septo-optic dysplasia.
Similarly in case 3, the optic nerves are described as pale with one larger than the other. This does not make for a diagnosis of ONH. Size of the optic nerve is very hard to determine, especially in very small children and infants. Photos are needed. This patient also has a multitude of systemic malformations, atypical for associations with ONH.
Response: Unfortunately, no images were taken during the exam in the NICU and the child died shortly thereafter. We reviewed the ophthalmology consult note and added details to the manuscript about abnormal vasculature around the optic nerves that was seen in addition to the pallor and asymmetry. He was given a clinical diagnosis of septo-optic dysplasia by the Genetics team that evaluated him at the time.
I am not sure that the observation/pearl that an ataxic gate requiring the use of a walker is a clue to the presence of SOX2 variants in patients with SOD. This is only one case and this is a common occurrence in patients with neurological deficits and associated brain dysfunction and malformations.
Response: We removed the focus on ataxia.
In summary, while the pathogenicity of the sequence variants has been well supported, the phenotypes they are associated with have in common some neurological malformations such as absence of the septum pellucid or corpus callosum. This reviewer is not convinced that the term SOD can be applied because of the lack of ophthalmoscopic evidence that they have ONH. Maybe the text could be rewritten acknowledging the variability of the phenotypes (as expected) and the difficulty in diagnosing classical SOD. Photographs of the optic nerves would add significantly to the paper.
Response: We agree that the phenotypic variability complicates the diagnosis of SOD. We have added text emphasizing the variability, the fact that all individuals in this study had a clinical diagnosis of SOD and acknowledging the less typical optic nerve findings in the individuals with genetic diagnosis.
Reviewer 2 Report
This study aimed to focus on genetic diagnoses in Septo-optic Dysplasia (SOD). The authors collected and analyzed six families with SOD using whole exome sequencing and identified de novo variants in SOX2, SHH, and ARID1A. This is a study with positive and future application. Authors clearly explained the background, M&M and results. Based on the novel findings obtained from this present clinical study, it is highly possible that others can collect more clinical samples to extend this study and gain more solid data to support these observation. Writing and format presentation seems suitable.
Author Response
Reviewer comment: This study aimed to focus on genetic diagnoses in Septo-optic Dysplasia (SOD). The authors collected and analyzed six families with SOD using whole exome sequencing and identified de novo variants in SOX2, SHH, and ARID1A. This is a study with positive and future application. Authors clearly explained the background, M&M and results. Based on the novel findings obtained from this present clinical study, it is highly possible that others can collect more clinical samples to extend this study and gain more solid data to support these observation. Writing and format presentation seems suitable.
Response: Thank you!
Round 2
Reviewer 1 Report
The authors have done a great job in responding to my comments. I note the addition of the presence of a single incisor in case 2. This finding is interesting and points to the presence of pituitary or midline brain defects and Indeed a possiboe SHH mutation.
I have three additional recommendation to improve the revised manuscript:
1. In the introduction the authors should define the diagnostic criteria for SOD.
2. A discussion of the significance of the single incisor in case 2, especially that the sequence variant in this case has been previously described.
3. The discussion could include some discussion about the variability in the type of optic nerve and ocular malformations in the setting of SOD